# Fear of COVID-19 among Healthcare Workers: The Role of Neuroticism and Fearful Attachment

**DOI:** 10.3390/jcm10194358

**Published:** 2021-09-24

**Authors:** Alfonso Troisi, Roberta Croce Nanni, Alessandra Riconi, Valeria Carola, David Di Cave

**Affiliations:** 1Department of Systems Medicine, University of Rome Tor Vergata, Via Montpellier 1, 00133 Rome, Italy; 2Studi Medici Mazzini, Viale Angelico 39, 00195 Rome, Italy; robertacrocenanni@libero.it; 3Psychiatry Residency Program, Medical School, University of Rome Tor Vergata, Via Montpellier 1, 00133 Rome, Italy; alessandrariconi@gmail.com; 4Department of Dynamic and Clinical Psychology, University of Rome La Sapienza, 00143 Rome, Italy; valeria.carola@uniroma1.it; 5Department of Clinical Sciences and Translational Medicine, University of Rome Tor Vergata, Via Montpellier 1, 00133 Rome, Italy; dicave@uniroma2.it

**Keywords:** COVID-19, fear, healthcare workers, neuroticism, insecure attachment

## Abstract

Fear of becoming infected is an important factor of the complex suite of emotional reactions triggered by the COVID-19 pandemic. Among healthcare workers (HWs), fear of infection can put at risk their psychological well-being and occupational efficiency. The aim of this study was to analyze the role of personality (i.e., the big five traits) and adult attachment in predicting levels of fear (as measured by the FCV-19S) in 101 HWs employed in a COVID-19 university hospital. The three significant predictors retained by the stepwise regression model were age (beta = 0.26, *t* = 2.89, *p* < 0.01), emotional stability (i.e., the inverse of neuroticism) (beta = −0.26, *t* = −2.89, *p* < 0.01), and fearful attachment (beta = 0.25, *t* = 2.75, *p* < 0.01). Older HWs with higher levels of neuroticism and fearful attachment reported more intense fear of COVID-19. Our results can be useful to identify vulnerable subgroups of HWs and to implement selective programs of prevention based on counseling and psychological support.

## 1. Introduction

COVID-19 has exposed healthcare workers (HWs) and their families to unprecedented levels of risk. While carrying out their duties, HWs face the occupational risk of being infected or unknowingly infecting others. Although HWs represent less than 3% of the population in the large majority of countries and less than 2% in almost all low- and middle-income countries, around 14% of COVID-19 cases reported to WHO is among HWs [1]. As of 1 June 2021, the number of coronavirus cases recorded among medical staff in Italy reached 135,054 [2].

Fear of becoming infected is an important factor of the complex suite of emotional reactions triggered by the COVID-19 pandemic [3]. There is a substantial difference between fear of infection and fear of noninfectious medical conditions (i.e., cancer, Alzheimer’s disease, heart disease, stroke, and diabetes) that are feared the most in ordinary times [4]. Fear of these degenerative diseases is largely cognitive and prompted by cultural inputs because their etiology and pathogenesis are largely dependent on risk factors and life habits that are typical of modern environments (e.g., extended longevity, high calories diet, sedentary lifestyle, obesity, smoking, drinking alcohol, pollution, etc.). By contrast, fear of infection is deeply rooted in our emotional brain because it reflects a psychological adaptation evolved to minimize the exposure to a wide and varying array of pathogens that were relatively common throughout the evolutionary history of *Homo sapiens* [5,6].

Studies conducted on the general population during the current pandemic showed that fear of COVID-19 is more intense among women and tends to increase with age [7,8,9,10]. Research on the psychological reactions to previous pandemics and epidemics suggests that increased levels of fear of infection are a risk factor for developing depression, hypochondriasis, and post-traumatic stress disorder [3]. A study of 256 adults in the United States found that fear of COVID-19 predicted both depressive symptoms and generalized anxiety [11]. Moreover, fear was also found to be strongly associated with other indicators of emotional distress, such as suicidal ideation, alcohol and drug use, and extreme hopelessness [12].

These findings from the general population gain even greater importance when applied to HWs. Fear of COVID-19 is an expected emotional reaction among HWs because the increased morbidity risk due to their occupational role adds to the natural fear of infection. Previous studies conducted during the current pandemic have confirmed the relevant incidence of fear of infection among HWs and the negative impact on their psychological well-being [13,14,15]. However, among HWs, fear of infection can put at risk their psychological well-being as well as their occupational efficiency. For example, frontline nurses with greater fear of COVID-19 report less job satisfaction and higher intent to leave the profession [16], and fear of infection has been shown to be a predictor of burnout [17]. Given the strong correlation between fear of infection and the development of negative psychological and occupational outcomes, an enhanced understanding of which HWs are more vulnerable has implications for the treatment and prevention of a broad range of pathologies (e.g., depression and post-traumatic stress disorder) and for the optimization of their professional performance. In addition, the identification of subgroups of HWs with greater levels of fear of infection can allow the implementation of personalized psychological support and programs to facilitate open communication [18,19].

The aim of the present exploratory cross-sectional study was to analyze the role of the big five personality traits and adult attachment style in predicting levels of fear of COVID-19 in a convenience sample of HWs employed in a COVID-19 university hospital. The rationale inspiring the choice of these individual variables was the large body of evidence showing the consistent association between personality traits, attachment style, and vulnerability or resilience to different types of stressful events [20,21,22,23], including stress response to COVID-19 pandemic [24,25]. In pre-COVID times, Taylor [26] predicted that individuals high in neuroticism are vulnerable to elevated distress during pandemics because they are sensitive to stress and threats of infection. His prediction has been confirmed by studies conducted during the current pandemic in the general population in the United States [11], Canada [24], and Italy [27]. Similar to neuroticism, insecure attachment has also been linked with enhanced stress sensitivity, emotional dysregulation, and propensity to experience negative affectivity [28,29]. Based on these previous studies, we hypothesized that higher levels of neuroticism and insecure attachment correlated with greater fear of COVID-19.

## 2. Materials and Methods

### 2.1. Participants

Participants were 101 healthcare professionals working in a major university hospital that was converted into a COVID hospital in spring 2020. Participants were recruited in the period between June and August 2020 by snowball sampling. In Italy, the COVID-19 pandemic was particularly invasive during the period between March and late April, then decreased in both the number of infections and in the seriousness of the illness throughout the summer of 2020 [30]. The study was conducted when vaccination was not yet available. Thus, all HWs attending the hospital (including the participants of this study) were obliged to adhere to the same strict preventive measures to reduce the risk of infection, independently of their professional roles. Participants’ mean age was 39.35 years (SD = 11.52, range: 21–70). In total, 64 were women and 67 were physicians. Other professional roles included nurses and laboratory technicians. Paper questionnaires were used to collect data. Participation was voluntary, and anonymity was guaranteed. To limit the selection biases of snowball sampling, we began with a set of initial informants that were as diverse as possible in terms of age, gender, and professional role followed by respondent-driven sampling method (i.e., weighting the sample in order to compensate for the initial non-random selection). Written informed consent was obtained prior to participation. The study was approved by the Ethical Committee of the Department of Dynamic and Clinical Psychology, Sapienza, University of Rome (Prot. n. 0000453 and Prot. n. 0000112).

### 2.2. Psychometric Measures

#### 2.2.1. Fear of COVID-19

Ahorsu et al. [31] have recently developed a brief and valid scale (FCV-19S) to capture an individual’s fear of COVID-19. The FCV-19S is a seven-item scale (e.g., “I am most afraid of COVID-19”, “My heart races or palpitates when I think about getting COVID-19”). The participants are asked to indicate their level of agreement with the statements using a five-item Likert-type scale. Answers included “strongly disagree”, “disagree”, “neither agree nor disagree”, “agree”, and “strongly agree”. The minimum score possible for each question is 1, and the maximum is 5. A total score is calculated by adding up each item score (ranging from 7 to 35). The higher the score, the greater is the fear of COVID-19. The Italian validation of the FCV-19S used in this study [32] showed robust psychometric properties (alpha = 0.82 and ICC = 0.72) and confirmed its stable unidimensional structure.

#### 2.2.2. Big Five Personality Traits

The Ten-Item Personality Inventory (TIPI) [33] is a short scale developed to measure personality traits according to the big five models (also known as the OCEAN model: openness to experience, conscientiousness, extraversion, agreeableness, neuroticism) in working or clinical settings in which assessment time is limited. The TIPI was developed using descriptors from other well-established big five instruments. Each of the 10 items is rated on a 7-point scale ranging from 1 (strongly disagree) to 7 (strongly agree). The version used in this study was the revised Italian version (I-TIPI-R) [34], which showed adequate factor structure, test–retest reliability, self-observer agreement, and convergent and discriminative validity with the Big Five Inventory (BFI). In the I-TIPI-R, the scale measuring neuroticism is inverted and named emotional stability (i.e., people scoring low on emotional stability have high levels of neuroticism). When reporting the results, we refer to emotional stability. Yet, in the discussion, to facilitate the comparison of our findings with those of previous studies, we refer to neuroticism.

#### 2.2.3. Attachment Style

To measure adult attachment style, we used the Italian version [35] of the Relationship Questionnaire (RQ) [36]. The RQ is a single-item measure made up of four short paragraphs, each describing a prototypical attachment pattern as it applies in close adult peer relationships. Participants are asked to rate their degree of correspondence to each prototype on a 7-point scale. The four attachment patterns (i.e., secure, preoccupied, fearful, and dismissing) are defined in terms of two dimensions: anxiety (i.e., a strong need for care and attention from attachment figures coupled with a pervasive uncertainty about the willingness of attachment figures to respond to such needs) and avoidance (i.e., discomfort with psychological intimacy and the desire to maintain psychological independence). The preoccupied, fearful, and dismissing patterns reflect different forms of insecure attachment.

The reliability estimates for the RQ self-ratings are comparable to those for other short questionnaires assessing adult attachment styles (test–retest *r*’s around 0.50) [37]. The RQ shows convergent validity with interview ratings of adult attachment [36]. As for discriminant validity, several studies have demonstrated that the RQ explains individual differences in cognition, emotions, and behaviors even after controlling for the big five personality traits [38].

### 2.3. Statistical Analysis

Statistical analysis was performed on a personal computer using SPSS for Windows, version 25.0 (SPSS, Inc., Chicago, IL, USA). Spearman’s rho was used to calculate bivariate correlations. Stepwise multiple regression analysis was used to identify significant predictors of infection fear. Although the primary aim of our study was to focus on personality traits and attachment as predictors of fear, in the first step of the stepwise multiple regression analysis, we included age, gender, and professional role to control for their possible confounding effects. There were no violations of the assumptions required by multiple regression. In particular, we used the Durbin–Watson statistic (value = 1.481) to check that the values of the residuals were independent, and variation inflation factors (VIF) scores (ranging from 1.012 to 1.033) and tolerances scores (ranging from 0.968 to 0.988) to check that there was no multicollinearity among the independent variables. The software G*Power 3.1.9.7 was used to calculate the minimum sample size for multivariate analysis.

## 3. Results

Table 1 reports the psychometric data for the entire sample. High levels of fear of infection (FCV-19S score > 18) were reported by 18% of the participants.

Nonparametric bivariate correlations between the I-TIPI-R, the RQ, and the FCV-19S showed higher levels of fear of COVID-19 in participants scoring lower on emotional stability (rho = −0.32, *p* < 0.01) and higher on preoccupied attachment (rho = 0.28, *p* < 0.01) and fearful attachment (rho = 0.27, *p* < 0.01). A stepwise multiple regression was conducted to determine which individual variables were the best predictors of fear of COVID-19, as measured by the FCV-19S. At step 1 of the analysis, age, gender (women vs. men), and professional role (medical doctors vs. other HWs) were entered into the regression model to control for their possible confounding effects. At step 2, the big five dimensions (i.e., extraversion, agreeableness, conscientiousness, emotional stability, and openness to experiences), as measured by the I-TIPI-R scores, were entered into the regression model. In step 3, the RQ scores for the four attachment patterns (i.e., secure, preoccupied, fearful, and dismissing) were entered into the regression model.

The final model explained 24% (R^2^) of the variance in the FCV-19S scores. The three significant predictors retained by the final model were age (beta = 0.26, *t* = 2.89, *p* < 0.01), I-TIPI-R emotional stability (beta = −0.26, *t* = −2.89, *p* < 0.01), and RQ fearful attachment (beta = 0.25, *t* = 2.75, *p* < 0.01) (Table 2). Older HWs with lower levels of emotional stability (i.e., higher levels of neuroticism) and higher levels of fearful attachment reported more intense fear of COVID-19.

## 4. Discussion

We found that older age predicted greater fear of infection. One possible explanation is that older HWs knew that they were at higher risk of critical COVID-19 symptoms [39]. In contrast, we found no correlations between gender, professional role, and fear of infection. It is likely that such missing correlations were idiosyncratic to our sample because a recent systematic review of 55 articles found that being a nurse and being female appeared to confer greater risk in terms of fear of infection [14].

We found that two personality traits, neuroticism and fearful attachment, were independent predictors of fear of infection. Neuroticism is a personality trait originally defined to include anxiety, emotional instability, worry, tension, and self-pity. This negative affectivity is accompanied by a pervasive perception that the world is a dangerous and threatening place, along with beliefs about one’s inability to manage or cope with challenging events [40]. In accordance with previous studies [11,24,27], our findings confirm the prediction by Taylor [26] that individuals high in neuroticism are vulnerable to elevated distress during pandemics because they are sensitive to stress and threats of infection. The original contribution of our study is that neuroticism is associated with a specific facet of emotional distress (i.e., fear of infection) and that such an association can be found among HWs.

The pattern of insecure attachment that emerged as a significant predictor of fear of COVID-19 over and above the effect of neuroticism was fearful attachment. The finding that fearful attachment was a significant predictor independent of neuroticism was expected because previous studies showed that correlations between attachment patterns and the big five personality traits are weak [41]. The RQ paragraph describing fearful attachment reads as follows: “I am uncomfortable getting close to others. I want emotionally close relationships, but I find it difficult to trust others completely or to depend on them. I worry that I will be hurt if I allow myself to become too close to others”.

We hypothesize that the psychological mechanisms linking fearful attachment with fear of infection are mainly related to dysfunctional coping strategies. In general, people with secure attachment tend to appraise stressful events in less threatening ways and to appraise themselves as able to cope effectively. In contrast, insecure attachment (including the fearful pattern) is associated with distress-intensifying appraisals (i.e., appraising threats as extreme and coping resources as deficient). Among people with fearful attachment, there is an additional factor that may increase fear of infection. They have a pervasive uncertainty about the willingness of significant others to respond to their needs for emotional support. Their typical discomfort with psychological intimacy and preference for emotional distance preclude self-disclosure, promote social avoidance, and can also work against attendance at support programs.

The few studies that have analyzed the relationship between adult attachment style and emotional reaction to the COVID-19 pandemic do not allow the assessment of the validity of our hypothesis. The theoretical paper by Rajkumar [42] makes no specific prediction about the pattern of insecure attachment that is expected to correlate with increased stress sensitivity to the COVID-19 outbreak. The report by Moccia et al. [43] on the Italian general population used the Attachment Style Questionnaire (ASQ) which, unlike the RQ used in the present study, does not measure the dimension of fearful attachment. Finally, the study by Lozano and Fraley [44] focused on sentinel behavior (only indirectly related to fear of infection) and found that people higher in attachment avoidance were less likely to protect themselves and protect others. We need further research to ascertain how different patterns of insecure attachment are associated with stress and coping during the current pandemic.

Based on the findings of the present study, neuroticism and fearful attachment may be viewed as vulnerability traits because of their link with fear of infection and the associated higher risk of developing stress-related psychiatric conditions. However, it is worth noting that fear of infection evolved as an adaptation to reduce the risk of contracting deadly diseases and that bold personality traits and lack of fear can lead to underestimating the risk of COVID-19 infection and eluding containment measures [45,46]. It is likely that the most adaptive emotional response to infection risk is to experience intermediate levels of fear (neither too high nor too low).

The main limitations of this study are related to the sampling method. By using snowball sampling, we had no information on how many HWs were approached and declined to participate. In addition, although the inclusion of HWs with different occupations and working in different wards provided a more complete picture of the impact of the pandemic, the limited number of participants and the variety of their duties limit the generalizability of our findings.

## 5. Conclusions

If confirmed by future studies based on larger samples, our results are relevant for policymakers and mental health professionals engaged to preserve HWs’ well-being and professional efficiency. The psychometric battery used in this study includes brief self-report scales that are easy to complete and useful to predict which HWs will be more inclined to react fearfully toward the COVID-19 outbreak. The identification of vulnerable subgroups would allow the selective implementation of prevention programs based on counseling and psychological support [18,19]. 

## Figures and Tables

**Table 1 jcm-10-04358-t001:** Psychometric scores for the entire sample (*n* = 101).

	Mean	SD	Range
FCV-19S	12.89	4.77	7–29
I-TIPI-R EXT	4.32	1.31	1–7
I-TIPI-R AGR	5.33	1.06	2.5–7
I-TIPI-R CON	5.72	1.27	1–7
I-TIPI-R EMS	4.78	1.28	1.5–7
I-TIPI-R OPE	4.84	1.30	1.5–7
RQ SECURE	4.20	1.67	1–7
RQ PREOCCUPIED	2.66	1.47	1–6
RQ FEARFUL	2.78	1.71	1–7
RQ DISMISSING	3.21	1.75	1–7

Legend: FCV-19S, Fear of COVID-19 scale; I-TIPI- R, Ten-Item Personality Inventory, Revised Italian Version; EXT, extraversion; AGR, agreeableness; CON, conscientiousness; EMS, emotional stability (the inverse of neuroticism); OPE, openness to experiences; RQ, Relationship Questionnaire.

**Table 2 jcm-10-04358-t002:** Results of stepwise regression analysis with fear of COVID-19 (FCV-19S) as the dependent variable, and sociodemographic data (step 1), big five personality traits (I-TIPI-R) (step 2), and adult attachment style (RQ) (step 3) as independent variables.

			FCV-19S	
		β	t	*p*
Step 1	Age	0.30	3.13	<0.01
	**Model**	**R^2^ = 0.09**	**F = 9.79**	**<0.01**
Step 2	Age	0.27	2.98	<0.01
	I-TIPI-R EMS	−0.30	−3.25	<0.01
	**Model**	**ΔR^2^ = 0.09**	**ΔF = 10.59**	**<0.01**
Step 3	Age	0.26	2.89	<0.01
	I-TIPI-R EMS	−0.26	−2.89	<0.01
	RQ FEARFUL	0.25	2.75	<0.01
	**Model**	**ΔR^2^ = 0.06**	**ΔF = 7.56**	**<0.01**
		**R^2^ = 0.24**	**F = 10.11**	**<0.01**

Legend: FCV-19S, Fear of COVID-19 scale; I-TIPI- R, Ten-Item Personality Inventory, Revised Italian Version; EMS, emotional stability; RQ, Relationship Questionnaire.

## Data Availability

The data that support the findings of this study are available from the corresponding author upon reasonable request.

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
