# Peer review of "Fear of COVID-19 among Healthcare Workers: The Role of Neuroticism and Fearful Attachment"

_jcm, 2021, doi:10.3390/jcm10194358_

Round 1
Reviewer 1 Report
no comments
Reviewer 2 Report
Thanks for the changes and clarifications. I have no further comment. congratulations!
This manuscript is a resubmission of an earlier submission. The following is a list of the peer review reports and author responses from that submission.
Round 1
Reviewer 1 Report
1 . Since regressions are being applied in order to find predictive variables, I would like to see a new table after table 1 , or as an alternative to a table a description, of all the correlations (only the significant ones) found between TIPI’s and RQ’s variables with the variable FCV-19S. Note that the correlations to be calculated are likely to be Sperman's and not Pearson's, which require the variables to verify normality assumptions first. This request is to support the choice of variables used in the stepwise multiple regression.
2. Correct the double point in value 2..98 in table 2, in step 2, Age.
3. Since the study had its limitations and these are identified in the discussion (line 195) I suggest that in the introduction it should be mentioned that the study is an exploratory study (suggest in line 56: The aim of the present exploratory study).
Author Response
Thank you for your comments. Below you find the changes that we made. They are listed according to the sequence of points you raised in your review.
- At the beginning of the Results section, we now report the significant correlations (Spearman’s rho) between the FCV-19S, big five traits, and attachment patterns (lines 149-151).
- The double point in Table 2 (2..98) has been corrected.
- In the Introduction, we make it clear that our study was exploratory (line 66).
Reviewer 2 Report
It is an interesting research that contributes to understanding the role of fear among healthcare workers and this studies are relatively few
Introduction: It is a good point when authors explain the consequences of fear of COVID-19 among healthcare workers is important for several reasons
Materials and methods: it is ok but in my opinion it will be interesting add more about procedure which participants answer instruments (on internet or paper). Besides, is important explain how was COVID-19 situation in Italy in this time between June and August (deaths, infecctions.....)
Discussion: attachment is repeat in some time, please change it by a synonymous; the authors argue their results with the literature because they have written :"The few studies that have analyzed the relationship between adult attachment style and emotional reaction to COVID-19 pandemic do not allow to assess the validity of our hypothesis" and they have tried to explain these
Limitations: please add one more
In general, the paper has been written well. It is very important that authors had obteined an ethical approval.
Author Response
Thank you for your comments. Below you find the changes that we made. They are listed according to the sequence of points you raised in your review.
- In the Methods section, the modality of test administration (paper questionnaires) is now reported (line 86). We refer to a study that reported epidemiological data on COVID-19 pandemic in Italy during the period of data collection (lines 82-84).
- Unfortunately, there is no synonymous for “attachment”. Thus, we are forced to repeat the same word anytime we refer to such a theoretical construct.
- When discussing the hypothesis linking fearful attachment with increased fear of infection, we make it clear that previous studies yielded inconsistent findings and did not specify which type of insecure attachment is associated with an exaggerated emotional response (lines 198-207).
- We have added one more limitation related to sampling procedure (lines 215-219).
Reviewer 3 Report
This research is interesting and addresses the fear of COVID-19 among healthcare workers. More specifically, the authors study three potential predictors of COVID-19 related fear among healthcare professionals: Demographic data (age, gender, professional role), personality traits (all the five) and attachment style (four different attachment patterns).
However, several important points need to be presented, clarified or developed. I list these points below in order of appearance in the text.
ABSTRACT
It is stated the aim of the study is the link between fear of COVID-19 and personality traits. However, we understand later that regression models also include the link between fear of COVID-19 and attachment style. There is no reason why it should not appear in the abstract. The text lacks clarity. Attachment styles do not represent personality traits.
Also, do not use abbreviations in the abstract (I-TIPI-R, RQ). Or at least define what we are talking about. It's hard to follow. That shouldn't be the case in the article, especially not in the abstract. Please use the same word throughout the text to talk about the same thing. For example, you only talk about personality at the beginning of the abstract, then in the results suddenly you talk about emotional stability and fearful attachment. This is confusing. What exactly are we talking about? It goes all the way through the text.
INTRODUCTION
- Lines 38-40: “Fear of infectious disease is substantially different from that evoked by other medical conditions because it reflects a psychological adaptation evolved to minimize the exposure to a wide and varying array of pathogens that were relatively common throughout the evolutionary history of Homo sapiens”.
I don't understand ... Do you mean the fear of other types of disease is different from the fear specific to COVID-19? What does this information bring to your study? Has the fear of other diseases been studied before? If so, explain the research and how it relates to your study.
- Lines 42-43: “Such a prediction has been confirmed by studies of healthcare workers during the current pandemic [5, 6]”.
What prediction? not clear what are you talking about.
- Lines 43-44: “However, compared to the number of studies conducted on the general population [7-10], the studies focusing on fear of COVID-19 among healthcare workers are relatively few”.
What did these studies show in the general population? So what can you expect in your population of healthcare workers?
- Line 46: “Studying the causes, correlates, and consequences of fear of COVID-19”.
Your study is correlational. You can't talk about causes or consequences. At most, you can talk about links.
- Lines 49-51. Very interesting! Again, most likely these studies are correlational and therefore we cannot speak of cause or consequence. Conversely, couldn't high anxiety and / or depression baseline levels be linked to higher fears about COVID-19? Healthcare workers who are already anxious / depressed could then have more fears about COVID-19. The literature on the link between anxiety and personality traits as well as on the link between anxiety and attachment is extensive. It would be interesting to talk about these studies. For example, an insecure attachment type has been linked to higher anxiety levels. You could therefore hypothesize an insecure attachment type will be linked to more fears about COVID-19.
- Lines 57-58: “We measured the Big Five personality traits and adult attachment style”
This information should appear in the method section, not in the introduction, unless your article was about these measuring instruments.
- Lines 61-62: “Based on these previous studies, we hypothesized that higher levels of neuroticism and insecure attachment correlated with greater fear of COVID-19”.
What previous studies? You don't present any study about neuroticism and attachment ... What are your assumptions based on? We don't even understand why you suddenly start talking about neuroticism.
MATERIAL AND METHODS
Participants
- I saw no report on how many healthcare professionals were approached about participating and how many were eligible, relative to the 101 who participated.
- Why not have recruited the participants directly in the targeted hospital (using posters or e-mail, etc.) rather than choosing the snowball sampling method, which comes with numerous selection bias for quantitative studies.
- Did you determine in advance how many participants you needed to adequately answer your research questions?
Statistics
Again, you talk about “the contribution of personality traits but not about the contribution of attachment styles. See my abstract comments.
DISCUSSION
- Lines 151-155: I would be interested to know if fears and anxiety increase with age in the general population, even without COVID-19? One explanation for this result could be the natural increase in anxiety with age.
- Line 155: Why include gender in demographic variables? Did you expect differences depending on gender? Does the literature tell us we should observe a difference between men and women?
- Many of the studies presented in the discussion should be in the introduction. We finally understand the literature that introduces the research subject by reading the discussion. The discussion should focus on the study results, not the literature.
Author Response
- Throughout the text, we have eliminated any ambiguity by distinguishing attachment from personality traits.
- Abbreviations (not explained) have been eliminated from the Abstract.
- We explain why fear of infection is different from fear of those medical conditions that are most feared during ordinary times. The explanation is supported by two references that were not quoted in the original version (lines 36-44).
- Previous studies inspiring research hypothesis are summarized in the paragraph of the Introduction (lines 66-77) and the study predictions are clearly stated at the end of the paragraph.
- In the subsection “Participants”, we give more info on the snowball sampling and the procedure (i.e. respondent-driven sampling method) used to limit its intrinsic biases (lines 87-90). In addition, the lack of information on how many HWs were approached and declined to participate is listed among the limitations of the study (lines 215-216).
- The software G*Power 3.1.9.7 was used to determine the minimum sample size for multivariate analysis (line 141).
- Lines 45-46 of the Introduction report the findings of previous studies showing that fear of COVID-19 tends to be higher in women and older people.
- The Discussion has been completely reorganized. We start by summarizing the main findings of the study and we have moved some portions to the Introduction and the Methods sections.
Reviewer 4 Report
The main aim of the present study was to analyze the role of personality traits and attachment in predicting levels of fear of COVID-19.
I have some comments I hope to be useful to improve your paper:
1) The words healthcare workers is used 27 times in the MS, I suggest using an abreviation to (HW) if you find apropriate.
2) in lines 48 to 55 you talk about excessive fear during pandemics. I suggest adding a not about lack of fear.
3) My main concern is in the discussion. It should start stating the main findings. Discussion is confusing. For example in lines 151-153 you explain why you add age, gender, and professional role to control for confounding effects - i think this information reads better in methods and not in discussion. Lines 166-168 were you finaly state the main contribution should in my opinion, be said before.
In sum, I feel discussion needs to be rewriten
Author Response
- The HWs abbreviation is now used throughout the text to indicate healthcare workers.
- This is a good point. In the Discussion, we briefly discuss the risks related to lack of fear and refer to 2 studies showing that your insight is supported by empirical data (lines 208-214).
- The Discussion has been completely reorganized. We start by summarizing the main findings of the study and we have moved some portions to the Introduction and the Methods sections.